# Integrin α7 and Extracellular Matrix Laminin 211 Interaction Promotes Proliferation of Acute Myeloid Leukemia Cells and Is Associated with Granulocytic Sarcoma

**DOI:** 10.3390/cancers12020363

**Published:** 2020-02-05

**Authors:** Nobuhiko Kobayashi, Tsukasa Oda, Makiko Takizawa, Takuma Ishizaki, Norifumi Tsukamoto, Akihiko Yokohama, Hisashi Takei, Takayuki Saitoh, Hiroaki Shimizu, Kazuki Honma, Kei Kimura-Masuda, Yuko Kuroda, Rei Ishihara, Yuki Murakami, Hirokazu Murakami, Hiroshi Handa

**Affiliations:** 1Department of Hematology, Gunma University Graduate School of Medicine, Maebashi 371-8510, Japan; m15702046@gunma-u.ac.jp (N.K.); takizawm@gunma-u.ac.jp (M.T.); itakuma@gunma-u.ac.jp (T.I.); m14702054@gunma-u.ac.jp (H.T.); hiroakis@gunma-u.ac.jp (H.S.); 2Laboratory of Molecular Genetics, The Institute for Molecular and Cellular Regulation, Gunma University, Maebashi 371-8510, Japan; toda@gunma-u.ac.jp; 3Oncology Center, Gunma University Hospital, Maebashi 371-8510, Japan; tsukamoto@gunma-u.ac.jp; 4Blood Transfusion Service, Gunma University Hospital, Maebashi 371-8510, Japan; ayoko@gunma-u.ac.jp; 5Graduate school of Health Science, Gunma University, Maebashi 371-8510, Japan; tsaitoh@gunma-u.ac.jp (T.S.); m12203035@gunma-u.ac.jp (K.H.); m14711024@gunma-u.ac.jp (K.K.-M.); m14711022@gunma-u.ac.jp (Y.K.); m13203005@gunma-u.ac.jp (R.I.); m13203034@gunma-u.ac.jp (Y.M.); hmura@gunma-u.ac.jp (H.M.)

**Keywords:** acute myelogenous leukemia, granulocytic sarcoma, integrin, laminin, extracellular matrix

## Abstract

Acute myeloid leukemia (AML) with granulocytic sarcoma (GS) is characterized by poor prognosis; however, its underlying mechanism is unclear. Bone marrow samples from 64 AML patients (9 with GS and 55 without GS) together with AML cell lines PL21, THP1, HL60, Kasumi-1, and KG-1 were used to elucidate the pathology of AML with GS. RNA-Seq analyses were performed on samples from seven AML patients with or without GS. Gene set enrichment analyses revealed significantly upregulated candidates on the cell surface of the GS group. Expression of the adhesion integrin α7 (*ITGA7*) was significantly higher in the GS group, as seen by RT-qPCR (*p* = 0.00188) and immunohistochemistry of bone marrow formalin-fixed, paraffin-embedded (FFPE) specimens. Flow cytometry revealed enhanced proliferation of PL21 and THP1 cells containing surface *ITGA7* in the presence of laminin 211 and stimulated ERK phosphorylation; this effect was abrogated following *ITGA7* knockdown or ERK inhibition. Overall, high *ITGA7* expression was associated with poor patient survival (*p* = 0.0477). In summary, *ITGA7* is highly expressed in AML with GS, and its ligand (laminin 211) stimulates cell proliferation through ERK signaling. This is the first study demonstrating the role of integrin α7 and extracellular matrix interactions in AML cell proliferation and extramedullary disease development.

## 1. Introduction

Acute myelogenous leukemia (AML) is a hematopoietic malignancy, the prognosis of which remains poor despite intensive chemotherapy or allogeneic stem cell transplantation combined with advanced supportive care [1,2] The WHO classification, based primarily on genetic findings, has replaced the previous FAB classification, and WHO treatment strategy and outcome prediction are broadly accepted [3,4]. However, further predictors of prognosis are required to ensure an optimal treatment strategy for AML patients.

We previously reported on the poor prognosis of AML with granulocytic sarcoma (GS) at diagnosis [5,6]. GS is an extramedullary disease, whereby leukemic cells form a mass outside the bone marrow and often emerge at the time of relapse and/or progression of AML, as well as at diagnosis [6]. Although the precise mechanism responsible for development of GS has not been fully characterized, type IV collagenase has been found to promote invasion of AML cells throughout the basement membrane, leading to GS development [7]. Considering the prognostic value of GS, clarifying its underlying mechanism(s) may prove useful for identifying novel therapeutic targets as well as improving the prognosis.

Recently, increasing attention has been paid to examining the role that interactions between cancer cells and their microenvironment, including the extracellular matrix (ECM) and stroma cells, play in carcinogenesis [8,9,10,11,12]. Integrins are receptors present on the cell surface that react with the ECM. By mediating intracellular signaling, they are involved in cell proliferation, cell adhesion, and cytoskeleton formation [13]. Integrins form dimers of α and β subunits and transduce various intracellular signals [8,14]. The ECM laminins serve as ligands for integrins with isoform-specific affinities. For example, laminin 211 has relatively high affinity for integrin α7β1 and the ensuing interaction transduces intracellular signals [8], including those beginning with the phosphorylation of focal adhesion kinase (FAK) and leading to the mitogen-activated protein kinase (MAPK)/ERK pathway, as well as the pathway initiated by binding of caveolin to Shc [14,15,16]. Several studies have shown that integrin α4β1, designated as very late antigen-4 (VLA-4), is highly expressed in AML cells. Further, cellular signaling via integrin and ECM binding has been described as playing important roles in AML progression, and hence, in overall disease prognosis [17,18,19,20]. However, the structural interactions that occur between integrins and the extracellular matrix to promote AML development have not been fully elucidated [21,22,23,24,25,26,27]

This study investigated the role of integrin α7 (*ITGA7*) in AML cells, focusing on its interaction with the ECM. The results demonstrate that *ITGA7* expression is a prognostic predictor for AML and suggest a novel mechanism for AML progression.

## 2. Results

### 2.1. Comprehensive Gene Expression Analysis of AML Cells by RNA-Seq

To evaluate the differential expression of genes in AML with or without GS, we first performed comprehensive gene expression analysis of bone marrow specimens obtained from patients with AML with GS (n = 7) or without GS (n = 7), respectively (Appendix A). The RNA-Seq gene expression data of these two groups were analyzed by Cufflinks on Basespace supplied by Illumina. Gene set enrichment analysis (GSEA) revealed a significantly different expression of cell surface molecules compared with the control group (Figure 1a) [28]. Based on the GSEA data, we selected *ITGA7* because the interaction between this integrin on leukemic cells and the ECM has not yet been studied but is speculated to play a role, especially in GS where leukemic cells are surrounded by a microenvironment different from the bone marrow (Figure 1b). *ITGA7* gene expression in AML was confirmed by The Cancer Genome Atlas (TCGA) (Appendix A). The gene expression of integrin β1, which pairs integrin α subunits, was also confirmed by our data (Appendix A).

### 2.2. ITGA7 Is Expressed in Both AML Patients and Cell Lines

Next, we examined *ITGA7* expression in bone marrow samples from 64 AML patients (9 with GS and 55 without GS), whose demographics are summarized in Table 1. Reverse-transcription quantitative polymerase chain reaction (RT-qPCR) revealed that *ITGA7* expression was significantly higher in AML patients with GS compared with those without GS (*p* = 0.00188) (Figure 2a). *ITGA7* expression was also confirmed in the GS formalin-fixed, paraffin-embedded (FFPE) tissue sections (n = 5) (Figure 2b).

The cell membranes and nuclei of atypical leukemic cells were immunohistochemically stained in FFPE specimens of bone marrow clots and GS. Both were positive for integrin α7 in cases with high expression of *ITGA7* mRNA (Figure 2c,d). Flow cytometric analysis in AML samples confirmed the presence of integrin α7 on the cell surface (Appendix A).

In addition, *ITGA7* expression in three AML cell lines was determined for functional studies. Among the five cell lines tested, PL21, which was established from AML accompanied by mediastinal GS, expressed the highest level of *ITGA7*, THP1 showed a moderate expression, and HL60 the lowest (Figure 2e). These findings are very similar to the RNA-Seq data obtained from Genentech via the Expression Atlas (https://www.ebi.ac.uk/gxa/home) (Appendix A). In addition, the PL21 and THP1 cell lines expressed integrin α7 on the cell surface, whereas HL60, Kasumi-1, and KG-1 did not (Figure 2f and Appendix A).

### 2.3. Laminin 211 Stimulates ERK Phosphorylation in AML Cell Lines Expressing Integrin α7

We hypothesized that laminin stimulation, an integrin ligand, may enhance proliferation of AML cells via integrin α7β1. Based on previous reports regarding the affinity between laminin and integrin isoforms [29,30], we assessed the phosphorylation of intracellular proteins in the presence of different laminin isoforms. To determine whether laminin 211 effectively transduces intracellular signals, ERK1/2 phosphorylation was detected in PL21 and THP1 cells upon stimulation with laminin 211. The results showed that laminin 211 stimulation for 15–60 min caused a gradual increase in the pERK1/ERK1 and pERK2/ERK2 ratios in both PL21 and THP1 cells, reaching a maximum of nearly twice the initial concentration at 60 min (Figure 3a,b). Alternatively, in HL60 cells, no laminin-211-mediated change was observed in the pERK1/ERK1 ratio; however, the pERK2/ERK2 ratio was 2.2-fold higher following 15 min of stimulation (Figure 3c). Further, laminin 411 stimulation also mildly increased the pERK1/ERK1 and pERK2/ERK2 ratios in PL21 cells and the pERK2/ERK2 ratio in HL60 cells but not in THP1 cells (Figure 3a–c).

Based on these results, ERK inhibitor II or the Akt inhibitor Wortmannin were added to cells to determine if signaling through laminin 211 was involved in cell proliferation. Proliferation of PL21 cells was generally suppressed in the presence of these inhibitors, while that of THP1 cells was significantly suppressed (Figure 3d).

### 2.4. ECM Laminin 211 Promotes Proliferation of AML Cell Lines by Expressing Integrin α7

Next, based on the phosphorylation assay results, we evaluated the difference in growth rate and morphological changes in culture dishes covered with various laminin isoforms.

Laminin 211 significantly increased the proliferation rate of PL21 cells compared with both laminin 411 and control during 72 h of culture (laminin 211 vs. laminin 411: *p* = 0.012; laminin 211 vs. control: *p* = 0.012; Figure 4a). Similar results were obtained with the THP1 cell line, where laminin 211 increased the proliferation rate compared with both laminin 411 and control (laminin 211 vs. laminin 411: *p* = 0.023; laminin 211 vs. control: *p* = 0.012; Figure 4b). In contrast, in HL60, Kasumi-1, and KG-1 cells, which do not express integrin α7, laminin 211 did not increase the proliferation rate (laminin 211 vs. laminin 411: *p* = 0.16; laminin 211 vs. control: *p* = 1.0; Figure 4c and Appendix A). Laminin 411 did not affect the proliferation rate in any of these three cell lines when compared to the control, and no morphological changes were observed following stimulation with either laminin 211 or laminin 411. Further, an adhesion study did not show adherence of AML cell lines to laminin (Appendix A), while cell cycle analysis via BrdU/7-AAD and flow cytometry demonstrated only a slight increase in the number of cells in S phase and G2/M phase (Appendix A).

To confirm the role of integrin α7 on cell proliferation, integrin α7 was knocked down by shRNA in PL21 cells. The results showed a diminished stimulatory effect by laminin, although it should be noted that cell growth was also suppressed by integrin α7 knockdown (Figure 4d).

To confirm that the interaction between laminin and integrin α7 stimulated ERK and Akt signaling, thus explaining their involvement in leukemic cell proliferation, the addition of ERK and Akt inhibitors was assessed. Both inhibitors induced a slight suppression in cell growth. Specifically, the ERK inhibitor abrogated growth promoted by laminin 211 in PL21 and THP1 cells (Figure 3d). Additionally, immunostaining confirmed the localization of the laminin α1 subunit, which constitutes the laminin 211 ligand, in tissues around GS (Appendix A).

### 2.5. Clinical Implications of ITGA7 Expression in Bone Marrow AML Cells

Next, the overall survival (OS) and relapse-free survival (RFS) of AML patients categorized by *ITGA7* expression level were evaluated. Due to the small number of cases in this study, the cut-off value for *ITGA7* expression was set using the quartile method. Clinical prognoses were analyzed in relation to *ITGA7* expression as either below (low-expression group) or above (high-expression group) 25% of cases. Fifty-two patients who had received chemotherapy were included. Although the low-*ITGA7*-expression group (median not calculated) exhibited a more favorable median 5-year OS compared with the high-expression group (1.03 years) (*p* = 0.0047), no statistical difference was observed in the 3-year RFS between the low (1.63 years) and high (0.99 years) *ITGA7* groups (*p* = 0.182) (Figure 5A,B).

## 3. Discussion

In this study, we determined that the adhesion molecule integrin α7 was highly expressed in AML cells, especially in AML with GS. Its specific ligand, laminin 211, stimulates cell proliferation through the ERK signaling cascade, suggesting that the AML–ECM interaction is important for extramedullary disease formation.

Several factors and molecules associated with GS formation have previously been reported [31,32,33,34]. For example, type IV collagenase has been shown to facilitate the infiltration of the basement membrane and cause extramedullary invasion [7]. However, additional mechanisms and molecules are likely involved in GS formation. Comprehensive gene expression analysis via RNA-Seq is a widely accepted method for identifying novel candidate genes in an unbiased manner. GSEA based on our RNA-Seq data revealed that differential expression of cell surface molecules was key for GS generation. *ITGA7*, which encodes the adhesion molecule integrin α7, was selected since integrins have been described as playing an important role in cancer progression [35,36,37] as well as normal cell processes [38,39,40] through ECM binding [41,42] and transduction of cellular signaling [43,44,45]. Although the expression of *ITGA7* in glioblastomas correlates to poor prognosis [35], the clinical significance of *ITGA7* expression is controversial in other malignancies [35,36,37,46,47,48,49,50,51], and its role and expression in AML have not yet been characterized.

Our RT-qPCR results demonstrated significantly higher expression of *ITGA7* in AML with GS in a large number of patient samples. The expression of the cell surface integrin α7 protein was also confirmed by immunohistochemistry of AML and GS samples. These data suggest a role for integrin α7 in GS formation. Although we also found that *ITGA7* expression was markedly elevated in GS tissue, the RNA extracted from this tissue also included that of surrounding tissues which naturally express *ITGA7*; hence, the data showing elevated *ITGA7* in GS samples must be interpreted carefully. Furthermore, a significant difference in *ITGA7* distribution was noted in the non-GS-group tissues. One explanation for this may be that the high *ITGA7* expression contributed to occult extramedullary AML infiltration without apparent GS formation, because in this study, we grouped AML with GS by histological proof or certification with CT imaging based on medical records. However, since we did not investigate factors regulating *ITGA7* expression in this study, it is difficult to fully explain these results. Specifically, since platelet-derived growth factor (PDGF) is shown to upregulate both the mRNA and protein expression of *ITGA7* in vascular smooth muscle cells [52], PDGF may have also contributed to integrin expression in the current study.

Integrin α7 forms a dimer with the ubiquitous integrin β1 and activates intracellular signals using laminin as a ligand [53,54]. Integrin α7β1 is expressed in muscle and involved in the formation of muscle fibers through interaction with the ECM [38,39,40]. Laminin 211, a laminin isoform with a relatively high affinity for integrin α7β1, is expressed mainly in the skeletal and cardiac muscles, peripheral nerves, testes, thymus, and bone marrow [30,53,54,55,56]. It has also been reported that chronic inflammation stimulates growth of dormant cancer cells, linking the ECM to tumor development [57]. Thus, our data showing that proliferation of leukemic cells expressing integrin α7 is promoted by the ECM laminin 211 suggests that the growth advantage experienced in extramedullary microenvironments, such as muscles with abundant ECM, favors the generation of GS.

ERK phosphorylation following binding of laminin 211 to integrin α7 supports the idea that ECM–integrin interaction can stimulate an intracellular signal cascade in leukemic cells. ERK signal transduction plays an important role in leukemic cell proliferation and is often implicated in sensitivity and resistance to leukemia therapy [58]. Integrin α7 knockdown also significantly affected cell growth, confirming similar previous observations in glioblastoma cells [35] and reinforcing the importance of this integrin in AML cells. Further, AML has been correlated with increased expression of ECM-related genes and the degradation of surrounding ECM following release of matrix metalloproteinases [21,59]. Hence, AML cells appear to actively exploit the ECM to favor their growth and maintenance. The ECM–integrin interaction identified here may stimulate other signaling pathways with roles beyond cell proliferation, which will require further examination.

Finally, survival data demonstrated that low *ITGA7* expression was associated with improved OS, as the expression of *ITGA7* provides an advantage to GS when growing in the extramedullary site, thereby favoring its development. Additionally, *ITGA7* expression in bone marrow leukemic cells may impact survival through this advantage afforded to GS. However, in our previous and present studies, GS demonstrated clinical significance only in terms of RFS and not OS, suggesting an alternate role for integrin α7 in AML compared with its role in GS formation. We did not quantify integrin α7 protein in all AML patients in this study, and the clinical utility of doing so should be pursued in the future.

## 4. Materials and Methods

### 4.1. Patient Samples

A total 64 AML patients (9 with GS and 55 without) participated in this study. Prior to the collection of patient samples, approval was obtained from the Institutional Review Board of Gunma University Hospital (IRB code: #1295). Informed consent about use of clinical samples was obtained from all patients. Bone marrow aspiration was performed at the time of AML diagnosis. Mononuclear cells from the bone marrow of the patients were separated with a gradient medium and the cell pellet was resuspended in RPMI 1640 containing dimethyl sulfoxide (DMSO) as cell stock medium and stored at −80 °C.

### 4.2. Cell Culture

The AML cell lines used in the experiments were PL21, THP1, HL60, Kasumi-1, and KG-1. HL60 (#CCL240) was purchased from the American Type Culture Collection (ATCC, Manassas, VA, USA). PL21 (#JCRB1319), THP1 (#RCB1189), KG-1 (#JCRB0065), and Kasumi-1 (#JCRB1003) were from the Riken Cell Bank (Tsukuba, Japan). PL21 was derived from an AML M3 patient without t (15;17) with myeloid sarcoma at a mediastinal site. The THP1 cell line was derived from an AML M5 patient, whereas HL60 was from an AML M2 patient [60]. All cell lines were maintained in RPMI 1640 medium with 10% fetal bovine serum (FBS), 100 units/mL of penicillin, 100 μg/mL of streptomycin, and 0.25 μg/mL of amphotericin (#15240062; Thermo Fisher Scientific, Waltham, MA, USA) at 37 °C and 5% CO_2_.

### 4.3. Lentiviral Constructs and ITGA7 Knockdown

*ITGA7* knockdown was performed with a lentiviral transduction using a shRNA sequence. Lentivirus preparation was performed as previously described [61]. The *ITGA7* target sequence [35] was as follows: Forward, 5′-CCGGCCTCCGGGATTTGCTACCTTTCTCGAGAAAGGTAGCAAATCCCGGAGGTTTTT-3′; Reverse, 5′-AATTAAAAACCTCCGGGATTTGCTACCTTTCTCGAGAAAGGTAGCAAATCCCGGAGG-3′.

### 4.4. Cell Proliferation Assay

Human recombinant laminin (#BLA-LN211-02, #BLA-LN411-02; Biolamina, Sundbyberg, Sweden) was diluted to 10 μg/mL with Dulbecco’s phosphate-buffered saline (#14040141; Invitrogen, Carlsbad, CA, USA) according to the manufacturer’s protocol. Next, 70 μL of the diluted laminin was added to each well of a 96-well plate (#353872; Corning Inc., Corning, NY, USA) and stored at 4 °C overnight for coating. PL21, THP1, and HL60 cell numbers were adjusted to 1.5 × 10^5^ cells/mL with RPMI 1640 medium, and 100 μL (1.5 × 10^4^ cells/well) was transferred to each well after removal of the supernatant containing laminin. The cells were cultured at 37 °C and 5% CO_2_. For counting, 10 μL of the Cell Counting Kit-8 (#343-07623; Dojindo, Kumamoto, Japan) reagent was added to each well at various time points (0, 24, 48, and 72 h) and incubated at 37 °C with 5% CO_2_ for 2 h. After that, absorbance at 450 nm was measured using a Wallac ARVO-SX 1420 spectrophotometer (PerkinElmer, Waltham, MA, USA). The proliferation rate of cell lines was calculated by dividing the absorbance at 450 nm for each time point by the absorbance at 0 h.

Proliferation assays were also performed in the presence of ERK inhibitor II (#sc-203945; Santa Cruz Biotechnology, Dallas, TX, USA) and Wortmannin (#AG-CN2-0023-M001; AdipoGen Corporation, San Diego, CA, USA) diluted in DMSO. The final concentration of ERK inhibitor II was 10 μM (PL21) or 100 μM (THP1), whereas that of Wortmannin was 100 nM for both cell lines. DMSO’s final concentration was 1% in all cases. All experiments were performed in technical and biological triplicates.

### 4.5. RNA Extraction and RNA-Seq

Total RNA was extracted from bone marrow mononuclear cells using the RNeasy kit (#74104; QIAGEN, Hilden, Germany), or FFPE bone marrow clots from an AML M1 patient with GS or FFPE GS sections using the NucleoSpin total RNA FFPE XS kit (#U0969A; TaKaRa Bio, Shiga, Japan) following the manufacturer’s protocols. RNA purity and concentration were measured using a spectrophotometer (#B-80-3004-31; Implen GmbH, München, Germany). RNA integrity numbers were confirmed to be higher than seven using the Agilent RNA6000 Pico kit (#5067-1513) in an Agilent 2100 Bioanalyzer (Agilent, Santa Clara, CA, USA). RNA-Seq was performed on seven AML specimens with GS and seven without GS. To construct the RNA library, the NEBNext Ultra RNA Library Prep Kit for Illumina (#E7530S; NEB, Ipswich, MA, USA), NEBNext rRNA Depletion Kit (#E6310L), Agencourt AMPure XP (#A63881; Beckman Coulter, Brea, CA, USA), and Agencourt RNAClean XP (#A63987) were used following the respective manufacturer’s protocols. The quality and concentration of the amplified complementary DNA (cDNA) library were determined using the Agilent 2100 Bioanalyzer with the high-sensitivity DNA kit (#5067-4626; Agilent) and the EnSpire plate reader (#11310334; PerkinElmer). RNA-Seq was performed on a NextSeq 500 (Illumina, San Diego, CA, USA) with NextSeq 500/550 high-output kit v2 (#FC-404-2005; NEB). Raw data were aligned against the human genome reference (hg19) using TopHat version 2.0. Gene expression levels were quantified using Cufflinks & DE version 2.0. GSEA (Broad Institute, Cambridge, MA, USA) [62,63] was performed to determine the enrichment of specific gene sets in the RNA-Seq data. The RNA-Seq data were submitted to the DDBJ/EMBL/GenBank databases under accession number DRA009575.

### 4.6. RT-qPCR

Total RNA was first converted to cDNA with the high-capacity RNA to cDNA kit (#4387406; Applied Biosystems, Foster City, CA, USA) and then used for RT-qPCR to evaluate the relative expression of the *ITGA7* gene. The RT-qPCR reaction was measured on a 7300 Real-Time PCR System (Applied Biosystems) using the Power SYBR^®^ Green PCR Master Mix (#4367659; Applied Biosystems). Primer (TaKaRa Bio) sequences were as follows:

*ITGA7*: Forward, 5′-GCTGCCCACTCTACAGCTTTGAC-3′; Reverse, 5′-ACAATCACTTCCAGGGACTTCACA-3′;

β-actin: Forward, 5′-TGGCACCCAGCACAATGAA-3′; Reverse, 5′-CTAAGTCATAGTCCGCCTAGAAGCA-3′. The relative levels of *ITGA7* mRNA were evaluated by the 2^−ΔΔCt^ method by normalizing expression to the β-actin gene.

### 4.7. Flow Cytometric Analysis

Flow cytometric analysis was conducted using anti-human *ITGA7* antibodies (#ab195959; Abcam, Cambridge, UK). The cells were washed twice with cold phosphate-buffered saline (PBS) and incubated in PBS containing 2% FBS with either primary antibody or isotype, followed by fluorescein isothiocyanate (FITC)-conjugated secondary antibody. Samples were then analyzed on a BD FACS Canto II (Becton Dickinson, Franklin Lakes, NJ, USA), and data were analyzed using FlowJo software version 9.7.7 (FlowJo, Ashland, OR, USA).

### 4.8. Western Blot Analysis

Whole cell lysates were obtained using RIPA lysis buffer containing phenylmethylsulfonyl fluoride, protease inhibitor cocktail, and sodium orthovanadate (#sc-24948; Santa Cruz Biotechnology). Harvested cells were lysed in RIPA lysis buffer. Then, they were frozen and thawed twice and centrifuged at 15,000 rpm for 10 min. The supernatant was collected as whole cell lysate. The samples were suspended 1:1 in sample buffer and Laemmli 2× concentrate (#S3401-10VL; Sigma-Aldrich, St. Louis, MO, USA), and 5 μg of total protein per well was loaded onto an SDS-PAGE gel (4%–20% gradient gel, #4561095; Bio-Rad, Hercules, CA, USA). After resolving, the bands were transferred onto a polyvinylidene difluoride membrane (#IPVH00010; Merck Millipore Ltd., Billerica, MA, USA) for 1 h at 17 V and then blocked in Tris-buffered saline –0.5% Tween 20 (TBS-T) with 0.3% dry, nonfat milk for 30 min at room temperature. The blocked membrane was incubated with the primary antibody (1:1000, Erk1/2 monoclonal antibody, phosphor-ERK1/2 monoclonal antibody from the #12651 PDGF Receptor Activation Antibody Sampler Kit; Cell Signaling Technology, Danvers, MA, USA) at 4 °C overnight. After washing three times with TBS-T, the membrane was incubated for 1 h at room temperature with horseradish-peroxidase-conjugated secondary antibody (1:3000, #7074S, Cell Signaling Technology) and strep-tactin (1:10,000, #161-0381; Bio-Rad). Antibody-labeled proteins were detected by enhanced chemiluminescence with Signal Fire (#6883P3; Cell Signaling Technology) using the Image Quant LAS 4010 system (GE Healthcare, Chicago, IL, USA). Band intensities were measured with ImageJ software version. 1.51s.

### 4.9. Immunohistochemical Staining

Immunohistochemical staining was performed using 3′,3′-diaminobenzidine (DAB) according to the peroxidase complex method. Paraffin sections were dehydrated at 44 °C for 60 min and deparaffinized. The antigen was activated with citrate buffer at pH 9.0 (#415201; Nichirei, Tokyo, Japan) and blocked using 10% horse serum (#16050; Gibco, Gaithersburg, MD, USA) and 1% sodium azide. Sections were incubated with polyclonal rabbit anti-human *ITGA7* (1:400, #HPA008427; Abcam) at 4 °C overnight. The samples were then washed and stained using Histofine simple stain MAX PO (Nichirei) as the secondary antibody. Color development was acquired with DAB (#347-00904; Dojindo).

### 4.10. Statistical Analysis

All data were analyzed using EZR version 1.33 [64]. The RT-qPCR data were divided into control and test groups and analyzed by Student’s *t* test or the Mann–Whitney *U* test. For cell proliferation experiments, the growth rate at each time point was analyzed using the Friedman test. For OS and RFS calculations, cut-off values were determined by the quartile method and the relationship between *ITGA7* expression and presence of GS was derived. The log-rank test was applied for OS; RFS was determined by the clinical course. Differences were considered significant at *p* < 0.05. False discovery rates were generated using Cufflinks and GSEA for RNA-Seq data.

## 5. Conclusions

In conclusion, our data revealed that leukemic cells express the *ITGA7*-encoded integrin α7 in association with GS. The ECM laminin 211 functions as a ligand of *ITGA7* to stimulate ERK signaling and cell proliferation. This is the first study to demonstrate the interaction between integrin α7 and the ECM in leukemic cells. Further studies are needed to clarify the precise role of integrin α7 in leukemic cells and to develop novel therapeutic strategies targeting this molecule.

## Figures and Tables

**Figure 1 cancers-12-00363-f001:**
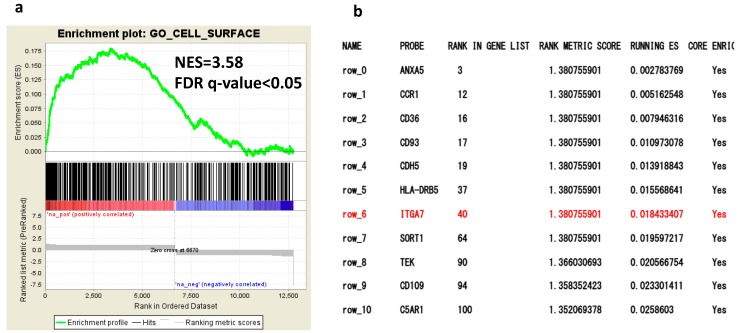
Gene expression in the acute myelogenous leukemia (AML) with granulocytic sarcoma (GS) group vs. AML without GS group. (**a**) Gene set enrichment analysis (GSEA) indicates that cell surface gene sets are enriched in AML with GS compared with AML without GS. Normalized enrichment scores (NES) and false discovery rate (FDR) *q*-values are given for the gene set. (**b**) List of genes enriched in the cell surface gene set derived from Gene Ontology annotations in GSEA (http://software.broadinstitute.org/gsea/msigdb/collection_details.jsp#C5) (Collection 5). The order of genes is ranked according to the running enrichment score.

**Figure 2 cancers-12-00363-f002:**
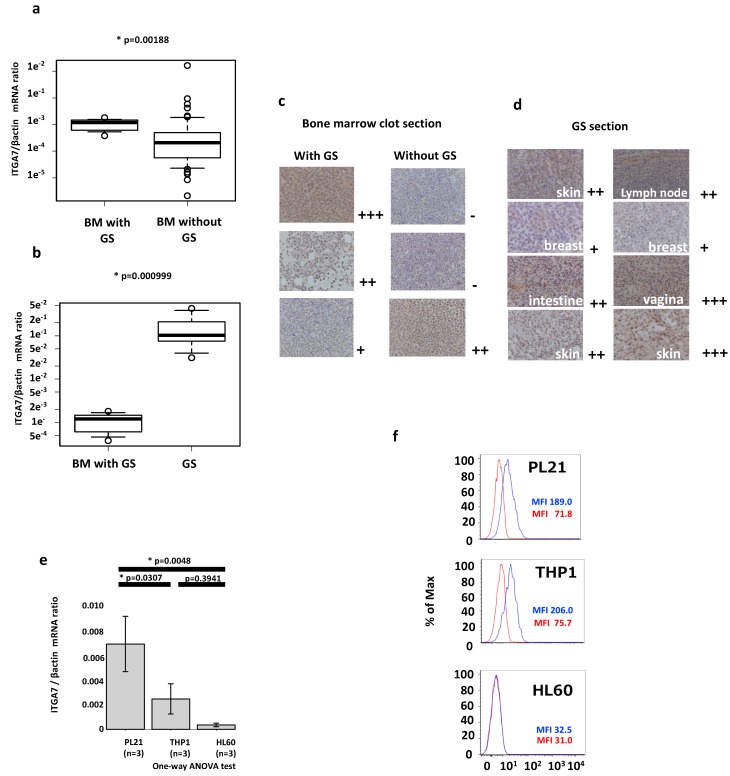
Validation of *ITGA7*/integrin α7 expression in patient samples and cell lines. (**a**) RT-qPCR-based expression of *ITGA7* in AML with GS and AML without GS. The *y* axis is logarithmic. (**b**) RT-qPCR-based expression of *ITGA7* in GS formalin-fixed, paraffin-embedded (FFPE) sections. The circle plots mean each expression data. The square shows box plot. (**c**) Expression of integrin α7 in bone marrow clots and (**d**) FFPE sections of GS. Immunohistochemical staining was positive in the nuclei, cell membrane, and cytosol of atypical cells in the GS section or bone marrow clots with GS. Staining intensity is semiquantitative and is expressed as + to +++. (**e**) RT-qPCR expression of *ITGA7* in AML cell lines. The vertical axis represents the *ITGA7*/β-actin mRNA ratio. The error bars represent standard deviation of the mean. (**f**) Flow cytometric analysis of PL21 and THP1 cells expressing integrin α7 on their cell surface (blue line). An isotype control was used for the primary antibody (red line). The vertical axis represents cell percentage. The horizontal axis represents arbitrary units of mean fluorescence intensity.

**Figure 3 cancers-12-00363-f003:**
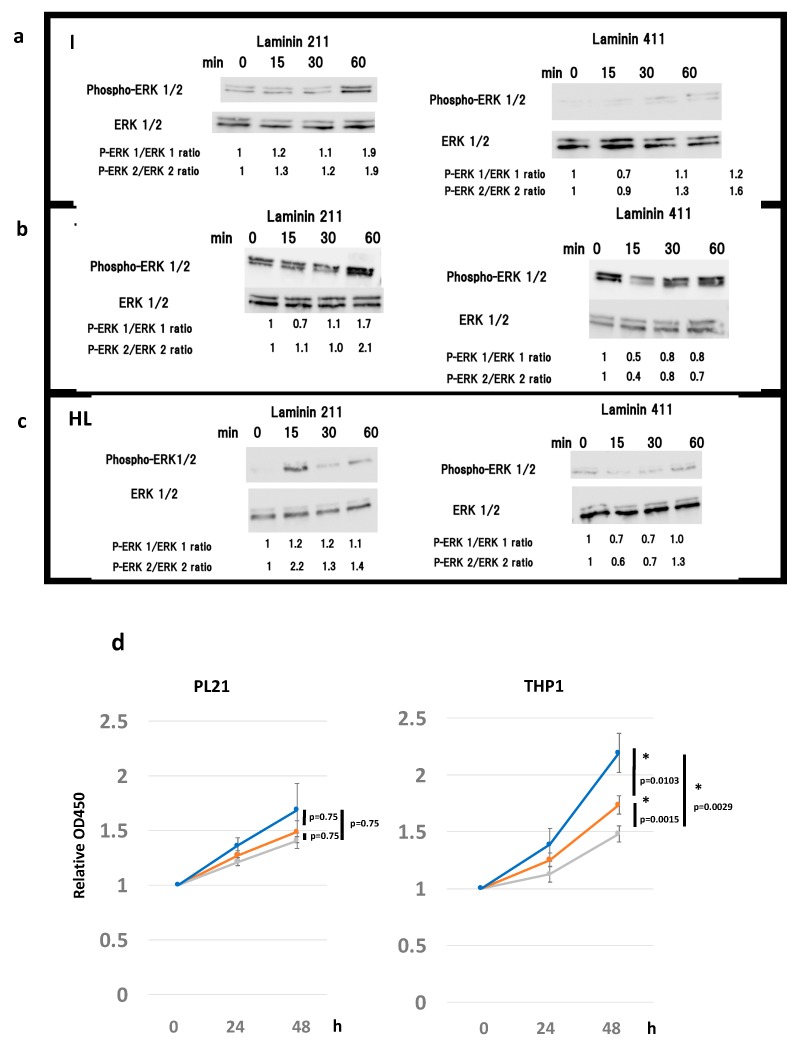
Effects of inhibitors on ERK1/2 and phospho-ERK1/2 protein expression and cellular proliferation indicate that laminin stimulates the integrin cascade. Western blot analysis of ERK1/2 and phospho-ERK1/2 after 15, 30, and 60 min of laminin 211, laminin 411, and control stimulation in (**a**) PL21, (**b**) THP1, and (**c**) HL60 cells. Numbers under the bands represent the relative band densities, as determined by densitometry, compared to the value at 0 min. (**d**) Proliferation of PL21 and THP1 cells in the presence of ERK inhibitor II (10 μM for PL21 and 100 μM for THP1, gray line), Wortmannin (10 nM, orange line), or dimethyl sulfoxide (DMSO) (control, blue line). The error bars represent the standard error of the mean. * *p* < 0.05 was considered statistically significant.

**Figure 4 cancers-12-00363-f004:**
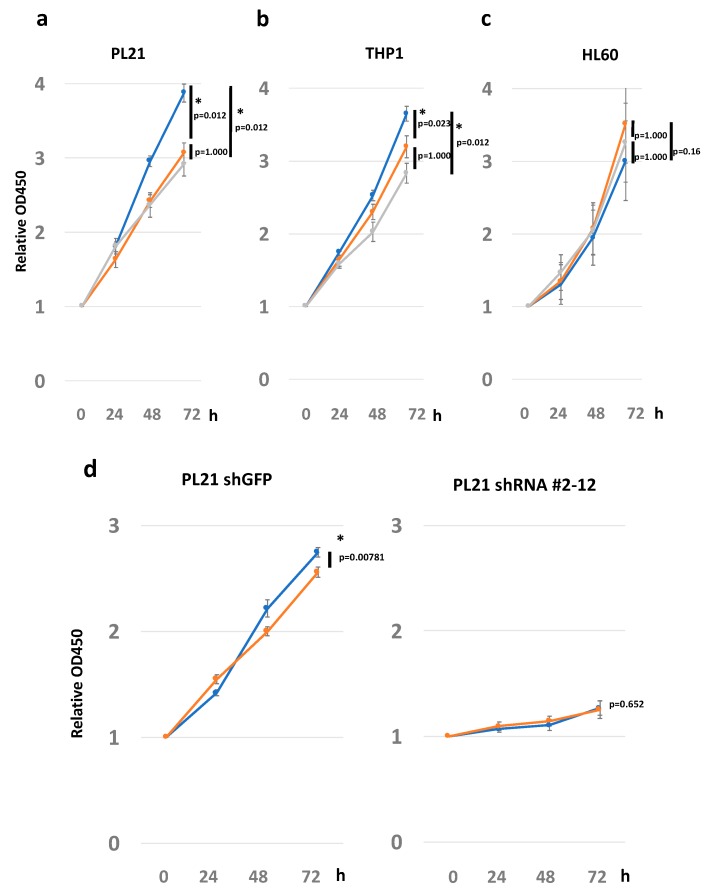
Cell proliferation assay with stimulation of laminin 211 and 411. Proliferation of (**a**) PL21, (**b**) THP1, and (**c**) HL60 cells in the presence of laminin 211, laminin 411, and the control groups. The vertical axis shows the proliferation rate starting at 0 h. The horizontal axis shows various time points. The experiments were replicated three times. The lines indicate proliferation on dishes coated with laminin 211 (blue), laminin 411 (orange), or the uncoated control (gray). (**d**) Proliferation of PL21 with shGFP (control) and sh*ITGA7* (#2-12). Lines represent proliferation on dishes coated with laminin 211 (blue) or laminin 411 (orange). Proliferation rates for each of the conditions were analyzed by the Friedman test or Wilcoxon’s rank sum test. The error bars show the standard error of the mean. * *p* < 0.05 was considered statistically significant.

**Figure 5 cancers-12-00363-f005:**
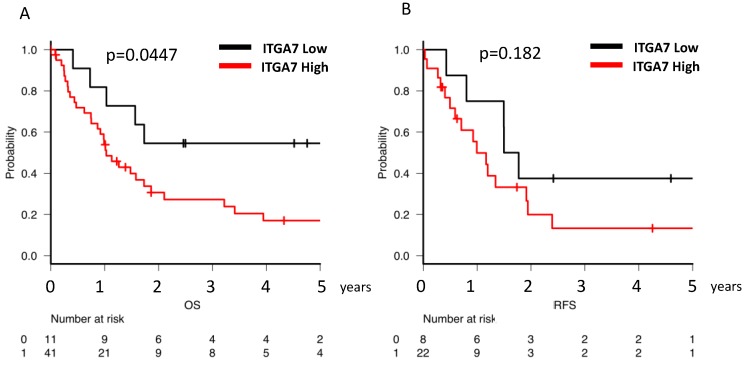
Clinical prognosis and *ITGA7* expression in AML patients. (**A**) Overall survival (OS) of patients with high and low *ITGA7* expression. (**B**) Relapse-free survival (RFS) of patients with high and low *ITGA7* expression. A cut-off value based on the quartile method was applied: the low-expression group represents fewer than 25% of cases, and the high-expression group includes the remaining cases.

**Table 1 cancers-12-00363-t001:** Patient characteristics of the 64 AML study participants.

		With GS	Without GS
Number of patients		9	55
Age (median)		59 (33–82)	65 (21–86)
White blood cells (μL)		14750 (2100–91,800)	4800 (700–205,800)
Blasts in peripheral blood (%)		46.0 (0–99)	18.0 (0–99)
Blasts in bone marrow (%)		56.3 (32–85.2)	49.0 (13–99)
Karyotype risk (%)	Favorable	1 (11.1)	8 (14.6)
	Intermediate	5 (56.6)	25 (45.4)
	Adverse	3 (33.3)	21 (38.1)
	N.A.	−	1 (1.8)
Chemotherapy	+	9 (100.0)	43 (78.1)
	−	0 (0.0)	8 (14.6)
	Unknown	−	4 (7.3)
Response	Complete Remission (CR)	7 (77.8)	20 (46.5)
	Non-CR	2 (22.2)	19 (44.1)
	Unknown	−	4 (9.3)

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
