# Peer review of "Integrin α7 and Extracellular Matrix Laminin 211 Interaction Promotes Proliferation of Acute Myeloid Leukemia Cells and Is Associated with Granulocytic Sarcoma"

_cancers, 2020, doi:10.3390/cancers12020363_

Round 1

Reviewer 1 Report

The authors have adequately addressed most of the comments, and the manuscript is significantly improved. The data correlating ITGA7 expression with improved overall survival is very interesting and adds interest to the paper.

However accurate citing of the literature is still lacking. The authors should discuss the following studies in either the introduction or discussion since they are very relevant to the current work in ITGA7. Specifically:

Identification of ECM component hyaluronan and CD44 interactions in primary human AML progression (Lin et al. Nature Medicine 2004) VLA-4 (Integrin alpha4 beta1) in AML growth and the integrin binding protein CD98 (Bajaj et al. Cancer Cell 2016) Integrin b1 signaling is important for AML progression (Miller et al. Cancer Cell 2013)

Reviewer 2 Report

All my concerns have been properly answered by the Authors. I feel the manuscript is now ready for publication and I advise it.

Reviewer 3 Report

This is interesting paper exploring the role of a specific integrin in the pathogenesis of granulocytic sarcoma, a form of extramedullary acute  myeloid leukemia. This paper has  some major and minor  deficiencies which must be corrected.

Major

1.  The abstract is misleading when it suggests 64 samples were tested... majority were non-GS  comparison cohort- this should be clarified.

2, Samples tested were all BM and no skin biopsies were include. This therefore tells only part of the story and would be more informative to include skin biopsies.

3.The methods say the RNA seq was done in patient whereas results say  they were done in 7 GS and 7 non-GS .... this is confusing and misleading.

4.The cell line PL21 is supposedly an AML M3 , a distinct disease which is separated from rest of the AML with a significantkly different prognosis and different treatment as well as outcome. This is therefore not a preferred  cell line for test for  a model of GS AML.

5. Why were other top ranking RNA seq outputs which are are also ECM proteins like ANXA5 and CCR1 not evaluted

 6. In fact in  fig 2 a, there is a significant distribution  differences in non-GS category which is not addressed in this paper

Minor

1. Significant English editing required. Simple edits like fonts etc not done.

2. The proliferation rate calculations and scale is not clear  in Fig 4

3 .Whilst GS AMLS are described as poor prognosis how do the  authors explain the high CR rates in their cohort , this appears significantly higher than  non GS.

4. What about the role of PDGF in expression of ITGA7 ... this not discussed at all.

5. Clinical implications are not addressed adequately. This paper focuses on GS AML whereas clinical implications are for all AML. If they wish to do  for all AML perhaps they should reattempt  this with online clinical trial data in a multivariate analysis.

Author Response

This manuscript is a resubmission of an earlier submission. The following is a list of the peer review reports and author responses from that submission.

Round 1

Reviewer 1 Report

In this study, Kobayashi et al. demonstrate an enhanced expression of Integrin alpha 7 in human AML patients with Granylocytic Sarcoma (GS) in comparison to AML without GS. They further show that Integrin alpha7 mediated interaction with Laminin promotes the proliferation of human AML cell lines, indicating that this interaction of leukemic cells with the ECM is important or cancer growth. While there is a critical need to better understand AML with GS, and the RNA-seq data presented is of significant clinical interest, the data presented in this paper does not conclusively prove a functional role for Integrin alpha 7 in AML (with GS) proliferation. Moreover, the authors claim that this is the first report of AML interaction with ECM is not accurate and should be reworded.

Specific comments:

1. There is only a modest increase in proliferation of THP-1 and PL21 cells in the presence of laminin (Figure 4). Does BrdU incorporation show more clear differences? Can the authors speculate why the HL-60 cell line does not show a difference and also provide data on the impact of growth and cell cycle kinetics of other AML cell lines (Kasumi, MV411 etc.)?

2. Since laminin can also interact with Dystroglycans, the data presented here does not clearly implicate a functional role for integrin alpha7 in AML proliferation. The authors should directly test the impact of blocking Integrin alpha7 on AML cell line growth using either shRNAs or function blocking antibodies.

3. Why is there a difference in response to Laminin-211 vs Laminin-411? What is the expression of these laminins in the bone marrow plugs and sections of AML with GS samples that have high Integrin alpha 7 staining (Figures 2 D and E)?

4. The authors claim in their abstract and in the text that this is the first work demonstrating a role for AML/ ECM in AML. Previous work has implicated the ECM component hyaluronan and CD44 interactions in primary human AML progression. While there is no report of a functional role for integrin alpha7beta1, integin alpha4beta1 (VLA-4) has been heavily implicated in AML growth. There are also reports of integrin b1 binding proteins (such as CD98) and integrin b3 in AML progression. These are only a few examples of the role of integrins and ECM interactions in AML. The authors should carefully review the literature and cite relevant work in their manuscript, and modify the abstract and text of their manuscript accordingly.

Reviewer 2 Report

There are several patients that have high expression of ITGA7 in without GS patients (Figure 2B). These “high ITGA7 patients without GS” is not included in Figure 2A and is misleading. Why did authors create two figures for Figure 2A and 2B although the same RT-PCR analysis is done for with GS and without GS? Moreover, as Figure 2B is showing several ITGA7 high patients and low patients, ITGA7 will not be the characteristics of with and without GS, which is critical for the conclusion of the manuscript.

What will the authors predict as the characteristics of “microenvironment different from the bone marrow” in GS? Will it still be the ECM or stromal cells, etc?

Figure 2D is trying to show different expression of ITGA7 between with GS and without GS. Although, it seems without GS also has ITGA7 expression and the result does not show difference.

Table 1 shows that WBC count and blasts in PB is high in “with GS”. Does author have any comment on this?

The authors state “Flow cytometric analysis in an AML sample confirmed the presence of cell surface ITGA7”, although the expression is relatively weak in figure S3. Can the authors show result of several patients that have more high expression of ITGA7?

The result of cell line; Why did HL60 have high expression of pERK2/ERK2 ratio after laminin 211 stimulation? This conflict with the authors result that laminin 211 promotes proliferation in cell lines with ITGA7 expression. There should be mechanism which laminin 211 induces proliferation other than ITGA7 stimulation. The authors should show the data or at least document this in the discussion section, as it will be misleading that laminin 211 induces proliferation via ITGA7 stimulation.

Line 159-160; “In contrast, the median RFS…” is not related to current study. The study is aiming to show difference between with GS and without GS, not karyotypes.

Line 161-163; The authors showed that RFS with GS is inferior, but failed to show that high ITGA7 expression is inferior. This means that expression of ITGA7 is not the cause of worse prognosis, and that it is not related to with or without GS. Can the authors show divided RFS combining with/without GS and ITGA7 low/high expression (showing four lines)?

Table 2 OS included 51 cases, while Table 3 RFS included 31 cases. Why did the authors not include 20 cases for OS? Please show the number of overlaps of low/high ITGA7 expression among with/without GS.

Minor remarks;

Figure 2F; isotype should be blue colored and Anti-ITGA7 should be red colored as in Suppl-Fig3.

Line 159; Fig. 5C should be 5D

Line 162; Fig. 5D should be 5C

Reviewer 3 Report

The manuscript by Kobayashi et al. is an interesting take on the biology of AML with Granulocytic Sarcoma (GS), a quite rare entity in the spectrum of AML presentations but obviously interesting for those who study the interactions between AML and the extracellular microenvironment.

The manuscript has evident strengths: a solid hold on clinical samples and data evaluation, a wide breath of techniques and a clear focus. I think it is overall a good manuscript worth of consideration, which could increase its robustness and be of more interest to the readership if the following suggestions are implemented.

MAJOR POINTS:

1)  RNA-seq data should be deposited (e.g., in Gene Expression Omnibus or another repository) to allow for future validation and re-use of the data.

2) GSEA results should be given as a whole (the various sets enriched in the two conditions should be all shown, e.g. in the Supplements) to provide readers with a clearer overall look. From there, Authors could go on with their experimental/analytical choice as they clearly indicate that they chose to pursue ITGA7 study out of scientific interest.

3) Figure S1 is supposed to deliver information on ITGA7 expression in TCGA LAML. Using the CBioPortal Oncoprint doesn’t really deliver this information, and Authors should show expression (based on FAB classes, if they prefer) as boxplots or similar (this can be easily done online using, e.g. the Xena browser).

4) Again in respect to TCGA LAML data, Authors decided to use the original NEJM 2013 data. Authors should notice that these data have been subsequently amended and expanded, so I would suggest that the Authors use either the final data available at GDC or that they “extract” LAML patients from the latest PanCancer cohort for the analysis.

5) Figure 2. Please provide quantification of staining intensities (and analysis) for images in D and E.

6) Figure 2. Please avoid cutting values from the axes in F, and rather show the values associated with them.

7) Though the Authors claim (line 128) that no morphological changes occur when AML cells are plated on laminins, adherence should be studied and reported as it might impact on WST data and thus on the interpretation of proliferation results.

8) The statement that “its specific ligand laminin 211 stimulates cell proliferation through ERK signaling” (Abstract, line 29) has no grounds. What these data show is that laminin 211 stimulates ERK phosphorylation and, concomitantly, proliferation. If such a claim is to be experimentally validated, Authors should at least repeat the proliferation experiments in the presence of ERK inhibitors.

9) ITGA7 is known to signal through both ERK and AKT (https://www.nature.com/articles/ncomms13568), the latter being overtly famous for its involvement in cell proliferation. Hence, the Authors should repeat proliferation experiments in the presence of Wortmannin or other Akt inhibitors to either show or rule out its involvement.

10) Dividing patients based on the maximization of an objective function (ROC curve in this case) is a sound method, though often very sensible to local minimas and maximas which may strongly perturb results when N is small (as in this case). I would suggest repeating univariate survival analysis dividing patients by quartiles, and comparing the 1st to the 4th quartiles.

MINOR POINTS:

1) Line 84: “To validate the results of ITGA7 expression determined by RNA-seq, we performed RT-qPCR on seven samples each from patients with and without GS, respectively.” should be rewritten as “To validate the results of ITGA7 expression determined by RNA-seq, we performed RT-qPCR on the same samples.” to avoid possible confusion with larger RT-qPCR results coming just after it.

2) A few more lines on the interaction between AML and the extracellular matrix, especially for what concerns the impact of the former on the latter (a topic of recent interest) should be developed to draw a clearer picture of the research scenario.